

# Non-standard neutrino interactions and neutral gauge bosons

**Julian Heeck[1,2]⋆, Manfred Lindner[3], Werner Rodejohann[3] and Stefan Vogl[3]**

**1** Service de Physique Théorique, Université Libre de Bruxelles,
Boulevard du Triomphe, CP225, 1050 Brussels, Belgium
**2** Department of Physics and Astronomy, University of California,
Irvine, CA 92697-4575, USA
**3** Max-Planck-Institut für Kernphysik, Saupfercheckweg 1, 69117 Heidelberg, Germany

⋆ julian.heeck@uci.edu

## Abstract

We investigate Non-Standard Neutrino Interactions (NSI) arising from a flavor-sensitive $Z'$ boson of a new $U(1)'$ symmetry. We compare the limits from neutrino oscillations, coherent elastic neutrino–nucleus scattering, and $Z'$ searches at different beam and collider experiments for a variety of straightforward anomaly-free $U(1)'$ models generated by linear combinations of $B - L$ and lepton-family-number differences $L_\alpha - L_\beta$. Depending on the flavor structure of those models it is easily possible to avoid NSI signals in long-baseline neutrino oscillation experiments or change the relative importance of the various experimental searches. We also point out that kinetic $Z$–$Z'$ mixing gives vanishing NSI in long-baseline experiments if a direct coupling between the $U(1)'$ gauge boson and matter is absent. In contrast, $Z$–$Z'$ mass mixing generates such NSI, which in turn means that there is a Higgs multiplet charged under both the Standard Model and the new $U(1)'$ symmetry.

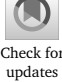

# 1 Introduction

The precision era of neutrino physics implies that small effects beyond the standard paradigm of three massive neutrinos may be detected. In particular new physics with a non-trivial flavor structure deserves careful consideration since it will modify neutrino oscillation probabilities in matter and may hinder our abilities to determine the unknown neutrino parameters at upcoming neutrino oscillation facilities, as discussed in Refs. [1–7]. The effects of Non-Standard neutrino Interactions (NSI) on low-energy observables are traditionally parametrized by an effective Lagrangian that describes couplings of neutrinos to quarks or electrons via [8–11]

$$\mathcal{L}_{\text{eff}} \propto \epsilon^f_{\alpha\beta} \left( \bar{\nu}_\alpha \gamma_\mu \nu_\beta \right) \left( \bar{f} \gamma^\mu f \right) \quad \text{with } f = e, u, d. \tag{1}$$

This effective interaction is clearly not $SU(2)_L \times U(1)_Y$ gauge invariant, begging the question how this Lagrangian is generated in a complete theory and what the mass scale of that theory is. The scale is of particular relevance for phenomenological studies since only processes with a momentum transfer smaller than the mass of the new physics can be described accurately by Eq. (1). Comparing NSI limits to other experimental data that probes much higher momentum transfers then typically requires a discussion of the full UV-complete theory. Several approaches have been followed in the literature to generate and study the interactions of Eq. (1) [12–21], here we discuss the origin of non-standard interactions in flavor-sensitive $U(1)'$ models [7,22–29]. The presence of additional Abelian symmetries is quite natural and can, for example, be motivated by Grand Unified Theories, string constructions, solutions to the hierarchy problem or extra dimensional models, see Ref. [30] for details and references.

We assume here the presence of a flavor-sensitive gauged $U(1)'$. In these theories the $Z'$ belonging to the $U(1)'$ is integrated out and generates the effective NSI Lagrangian Eq. (1).[1] Limits on the strength of the interaction can be translated into limits on the $Z'$ mass and gauge coupling. Those limits have to be compared with direct beam and collider searches, as well as neutrino–electron and elastic coherent neutrino–nucleus scattering results. In our discussion we will refer to the low-energy four-fermion operators and their impact on neutrino oscillations as NSI, while we discuss all observables with non-vanishing momentum transfer in terms of the high-energy $U(1)'$. This is the preferable notation for NSI mediated by rather light particles for which the effective NSI Lagrangian fails to describe all the relevant phenomenology.

The necessary ingredients for $Z'$-induced NSI are $Z'$ couplings to matter, i.e. electrons, protons or neutrons, as well as non-universal couplings to neutrinos. Neutrino oscillations would not be affected by flavor-*universal* NSI, $\epsilon \propto \mathbb{1}$, so NSI are actually a probe of *lepton non-universality*. This is interesting in view of the accumulating hints for lepton non-universality in $B$ meson decays (see Ref. [32] for a recent overview). While we will not attempt to make a direct connection between NSI and these tantalizing hints for new physics, it should be kept in mind as a motivation. The NSI model-building challenge is then to find realistic $U(1)'$ models with lepton non-universal $Z'$ couplings. As is well known, the classical Standard Model (SM) Lagrangian already contains the global symmetry $U(1)_B \times U(1)_{L_e} \times U(1)_{L_\mu} \times U(1)_{L_\tau}$ associated with conserved baryon and lepton numbers. A simple extension of the SM by three right-handed neutrinos – which are in any case useful to generate neutrino masses – allows one to promote $U(1)_{B-L} \times U(1)_{L_\mu-L_\tau} \times U(1)_{L_\mu-L_e}$ or any subgroup thereof to a local gauge symmetry [33]. We will focus on simple $U(1)_X$ subgroups, which are hence generated by

$$X = r_{BL}(B-L) + r_{\mu\tau}(L_\mu - L_\tau) + r_{\mu e}(L_\mu - L_e) \tag{2}$$

---

[1]The current–current structure of Eq. (1) for neutrino–quark scattering could also be induced by leptoquarks. The leptoquark Yukawa couplings automatically bring the desired lepton non-universality, but typically also lead to lepton-flavor and even baryon-number violation, which forces them to be very weakly coupled. While it is possible to eliminate some of the undesired couplings by means of a (flavor) symmetry [31], we will not pursue this direction here.

for arbitrary real coefficients $r_x$ [33] (see also Refs. [34–38]), potentially including $Z$–$Z'$ mixing. We stress that these $U(1)_X$ models are anomaly free and UV-complete, allowing us to reliably compare limits from NSI and other experiments. In their simplest form these models are also safe from proton decay and lepton flavor violation without the need for any fine-tuning, and can furthermore accommodate neutrino masses via a seesaw mechanism [33, 38]. This makes them perfect benchmark models for NSI, ideal to illustrate the importance of neutrino-oscillation limits compared to e.g. neutrino scattering constraints.

While $Z'$ bosons and NSI have been considered before [7, 22, 23, 25–27, 29], our work is distinct due to the following aspects: we stress the importance of whether the $Z'$ couples directly to matter particles (i.e. electrons, up- and down-quarks), or whether it couples to matter only via $Z$–$Z'$ mixing. We demonstrate that in the latter case $Z$–$Z'$ *mass mixing* is required to generate observable NSI in long-baseline oscillation experiments, implying non-trivial Higgs phenomenology. This is because mass mixing requires a Higgs multiplet which is charged under both the $U(1)'$ and SM gauge groups. Working with simple anomaly-free $U(1)'$ symmetries we furthermore stress the importance of the flavor structure of the underlying models, which strongly influences the size of the limits (via the sign of the generated $\epsilon$), as well as the importance of other constraints on the $Z'$ mass and gauge coupling. We also demonstrate that within simple UV-complete models it is possible to make terrestrial neutrino oscillation experiments insensitive to NSI, such that only scattering or collider limits apply.

The paper is organized as follows: In Section 2 we introduce the formalism of NSI and summarize current limits from neutrino oscillations. The interplay of the flavor structure of the $\epsilon$ is stressed by comparing COHERENT limits in different cases. Section 3 deals with the calculation of NSI operators when $Z'$ bosons are integrated out, with particular focus on whether kinetic or mass mixing is present. Specific examples from explicit models, which are anomaly-free when only right-handed neutrinos are introduced, are given. We conclude in Section 4.

## 2 Non-Standard Neutrino Interactions: Formalism and Limits

NSI relevant for neutrino propagation in matter are usually described by the effective Lagrangian

$$\mathcal{L}_{\text{eff}} = -2\sqrt{2} G_F \, \epsilon^{f\,X}_{\alpha\beta} \left( \bar{\nu}_\alpha \gamma_\mu P_L \nu_\beta \right) \left( \bar{f} \gamma^\mu P_X f \right), \tag{3}$$

where $X = L, R$ depends on the chirality of the interaction with $P_{L,R} = \frac{1}{2}(1 \mp \gamma_5)$ and $f \in \{e, u, d\}$ encodes the coupling to matter; $2\sqrt{2} G_F \simeq (174\,\text{GeV})^{-2}$ is a normalization factor that makes $\epsilon$ dimensionless. Relevant for neutrino oscillation experiments is only the vector part

$$\epsilon^f_{\alpha\beta} \equiv \epsilon^{f\,L}_{\alpha\beta} + \epsilon^{f\,R}_{\alpha\beta}, \tag{4}$$

because this induces coherent forward scattering of neutrinos in unpolarized matter. For non-trivial flavor structures, $\epsilon \not\propto \mathbb{1}$, this modifies neutrino propagation and oscillation in the Sun and Earth. In the following, we will denote this oscillation effect of the Lagrangian in Eq. (3) as NSI, in contrast to various other places where the Lagrangian and its UV-complete realization may show up. Limits on NSI parameters can be obtained by fitting neutrino oscillation data,

Table 1: $2\sigma$ bounds on the diagonal NSI $\epsilon^f_{\ell\ell} - \epsilon^f_{\mu\mu}$ assuming scattering on the fermions $f \in \{u, d, p, n, p+n\}$ from neutrino oscillation data assuming LMA, as derived in Ref. [40].

| $f$ | $\epsilon^f_{ee} - \epsilon^f_{\mu\mu}$ | $\epsilon^f_{\tau\tau} - \epsilon^f_{\mu\mu}$ |
|---|---|---|
| $u$ | $[-0.020, +0.456]$ | $[-0.005, +0.130]$ |
| $d$ | $[-0.027, +0.474]$ | $[-0.005, +0.095]$ |
| $p$ | $[-0.041, +1.312]$ | $[-0.015, +0.426]$ |
| $n$ | $[-0.114, +1.499]$ | $[-0.015, +0.222]$ |
| $p+n$ | $[-0.038, +0.707]$ | $[-0.008, +0.180]$ |

which is modified due to the additional Hermitian matter potential in flavor space

$$H_{\text{mat}} = \sqrt{2} G_F N_e(x) \begin{pmatrix} 1 + \epsilon_{ee}(x) & \epsilon_{e\mu}(x) & \epsilon_{e\tau}(x) \\ \epsilon^*_{e\mu}(x) & \epsilon_{\mu\mu}(x) & \epsilon_{\mu\tau}(x) \\ \epsilon^*_{e\tau}(x) & \epsilon^*_{\mu\tau}(x) & \epsilon_{\tau\tau}(x) \end{pmatrix}, \tag{5}$$

with normalized NSI $\epsilon_{\alpha\beta} = \sum_f \frac{N_f(x)}{N_e(x)} \epsilon^f_{\alpha\beta}$ and position-dependent fermion densities $N_f(x)$.[2] Since neutrino oscillations are not sensitive to a matter potential $H_{\text{mat}} \propto \mathbb{1}$, one can constrain only *two* diagonal entries, usually written in the form of differences as $\epsilon_{ee} - \epsilon_{\mu\mu}$ and $\epsilon_{\tau\tau} - \epsilon_{\mu\mu}$. Limits are typically obtained assuming a neutrino scattering only off one species $f \in \{e, u, d\}$. Recently, Ref. [40] has generalized this approach to allow for an arbitrary linear combination of up- and down-quark NSI, which in particular includes the case of scattering off protons ($f = p$: $\epsilon^p_{\alpha\beta} \equiv 2\epsilon^u_{\alpha\beta} + \epsilon^d_{\alpha\beta}$) or neutrons ($f = n$: $\epsilon^n_{\alpha\beta} \equiv \epsilon^u_{\alpha\beta} + 2\epsilon^d_{\alpha\beta}$). Limits on the diagonal NSI from oscillation data are given in Tab. 1, derived under the Large Mixing Angle (LMA) assumption for $\theta_{12}$ [40].[3] Three combinations will turn out to be of particular interest for our study: (i) $p+n$, (ii) $n$, and (iii) $p$. The combination $p+n$ corresponds to NSI couplings $-2\sqrt{2} G_F \epsilon^{p+n}_{\alpha\beta} (\bar{\nu}_\alpha \gamma_\mu P_L \nu_\beta) j^\mu_B$ to the baryon current

$$j^\mu_B = \frac{1}{3} \sum_q \bar{q} \gamma^\mu q \supset \bar{p} \gamma^\mu p + \bar{n} \gamma^\mu n, \tag{6}$$

from which we can obtain the relation with $\epsilon^{u,d}$ via $\epsilon^{p+n}_{\alpha\beta} \equiv (\epsilon^p_{\alpha\beta} + \epsilon^n_{\alpha\beta})/2 = (3\epsilon^u_{\alpha\beta} + 3\epsilon^d_{\alpha\beta})/2$. Pure neutron NSI are realized if the couplings to protons and electrons cancel in matter, a situation we will encounter for instance in Sec. 3.2. Pure coupling to protons, on the other hand, can under certain assumptions be used as a proxy for electron NSI.[4]

NSI mediated by a new neutral vector boson $Z'$ with coupling strength $g'$ and mass $M_{Z'}$ are generically of the form $\epsilon \sim (2\sqrt{2} G_F)^{-1} (g'/M_{Z'})^2$, even if the $Z'$ mass is tiny. The values of Tab. 1 then correspond to scales $M_{Z'}/g'$ from 140 GeV to 2.5 TeV, depending on $\alpha$, $\beta$, $f$, and

---

[2]Crossing through electrically neutral matter consisting of protons, neutrons and electrons, coherent forward scattering picks up NSI effects proportional to the number densities: $\epsilon^{\text{Matter}}_{\alpha\beta} = \epsilon^e_{\alpha\beta} + \epsilon^p_{\alpha\beta} + Y^{\text{Matter}}_n \epsilon^n_{\alpha\beta}$, where $Y^{\text{Matter}}_n = n_n/n_e$ is the ratio of neutron and electron number densities. For Earth matter, $Y^{\text{Earth}}_n = 1.051$ on average [39].

[3]See e.g. Refs. [5, 7] for recent discussions on the LMA-Dark solution.

[4] Limits on $\epsilon^p$ are not equivalent to $\epsilon^e$ despite the same electron and proton abundance in electrically neutral matter because they modify the neutrino detection process differently [40]. However, in the models considered in the following neutrino–electron scattering provides an independent constraint on the strength of the interaction which restricts the new-physics impact on the neutrino detection process in oscillations experiments such as Super-Kamiokande substantially. We stress that this is only an estimate and encourage a dedicated analysis of the interplay of $\epsilon^e$ and $\epsilon^q$. A summary of independent constraints on NSI from electrons $\epsilon^e_{\alpha\beta}$ which do not come from a global fit can be found in Ref. [11].

the sign of the coefficient. These have to be compared to limits from other processes, e.g. resonance searches for $Z'$ at the LHC or meson decays. Among the various processes which could be used to test a $Z'$, neutrino scattering off electrons [41,42] or nucleons [27] has the greatest similarity to NSI and the main difference between scattering experiments and NSI constraints is the momentum transfer: neutrino oscillations probe zero-momentum forward scattering and thus give limits on $M_{Z'}/g'$ that are independent of $M_{Z'}$ [25]. In contrast, the observations of neutrino scattering off quarks and electrons always requires a non-vanishing momentum transfer. Neutrino–electron scattering experiments are sensitive to $\mathcal{O}(1\,\text{MeV})$ momentum transfer while Coherent Elastic $\nu$–Nucleus Scattering (CE$\nu$NS), which has been measured by COHERENT [43] recently, currently allows to probe a momentum transfer $q$ of the order of $\sim 50\,\text{MeV}$. Future data from COHERENT and other experiments such as CONUS [44] will further improve this probe [7]. With initial neutrinos of flavor $\alpha$ (that is $\alpha = e$ for experiments with reactor neutrinos such as CONUS and $\alpha = e, \mu$ for experiments with pion beams such as COHERENT), the cross section for CE$\nu$NS on a nucleus $i$ with $Z_i$ protons and $N_i$ neutrons is proportional to the effective charge-squared

$$\tilde{Q}^2_{i,\alpha} \equiv \left[ N_i \left( -\frac{1}{2} + \epsilon^n_{\alpha\alpha} \right) + Z_i \left( \frac{1}{2} - 2s^2_W + \epsilon^p_{\alpha\alpha} \right) \right]^2 + \sum_{\beta \neq \alpha} \left[ N_i \epsilon^n_{\alpha\beta} + Z_i \epsilon^p_{\alpha\beta} \right]^2, \qquad (7)$$

assuming real NSI for simplicity. Due to the short neutrino propagation length one can neglect neutrino oscillations here. The COHERENT [43] experiment uses neutrinos from pion decay at rest, scattering on cesium and iodine, which leads to an expression for the number of CE$\nu$NS events

$$N_{\text{CE}\nu\text{NS}} \propto \sum_{i \in \{\text{Cs,I}\}} \left[ f_{\nu_e} \tilde{Q}^2_{i,e} + (f_{\nu_\mu} + f_{\bar{\nu}_\mu}) \tilde{Q}^2_{i,\mu} \right], \qquad (8)$$

with $f_{\nu_e} = 0.31$, $f_{\nu_\mu} = 0.19$, and $f_{\bar{\nu}_\mu} = 0.50$ as appropriate neutrino-flavor fractions for COHERENT. Note that experiments with reactor neutrinos such as CONUS are only sensitive to $\tilde{Q}^2_{i,e}$. CE$\nu$NS is obviously sensitive to different NSI combinations than oscillation data and therefore perfectly complementary. To assess NSI limits from COHERENT we follow Refs. [40,43,45] and construct a $\chi^2(\epsilon)$ function that is marginalized over systematic nuisance parameters.[5] Compared to oscillation-based limits on NSI, the limits from scattering experiments always imply a non-zero momentum exchange $q$, which has to be taken into account in NSI realizations with light mediators. Specifically for $Z'$ models, the above expression is only valid for $M_{Z'} \gg q \simeq 10\,\text{MeV}$, otherwise there is a suppression of the form $\epsilon \to \epsilon M^2_{Z'}/q^2$ [25]. In addition, neutrino scattering experiments are also sensitive to $\epsilon_{\alpha\beta} \propto \delta_{\alpha\beta}$ and are therefore invaluable as a probe of new flavor-*universal* interactions.

As examples we consider diagonal muon- and electron-neutrino NSI that come from scattering on baryons, i.e. $\epsilon^{p+n}$. Setting $\epsilon_{\tau\tau} = 0$ implies a strong bound from oscillation data due to the stringent constraint on $|\epsilon_{\tau\tau} - \epsilon_{\mu\mu}|$ (Tab. 1), so that COHERENT limits are weaker (Fig. 1 (left)). Setting on the other hand $\epsilon_{\tau\tau} = \epsilon_{\mu\mu}$ completely eliminates one of the two diagonal NSI constraints from oscillation data and thus renders COHERENT crucial to constrain the parameter space (Fig. 1 (right)). Although counterintuitive due to the absence of tau-neutrinos in the experiment, the COHERENT limits are particularly important for $\epsilon_{\tau\tau} \neq 0$, because this can weaken the strong oscillation constraints. As we will see in the following, COHERENT is indeed mainly relevant for simple $Z'$ models with $\epsilon_{\tau\tau} \sim \epsilon_{\mu\mu}$.

One lesson learned so far is that a possible underlying flavor structure of the $\epsilon_{\alpha\beta}$ strongly influences which experiment is most sensitive to them.

---

[5]See also Refs. [46–51] for discussions of NSI at coherent scattering experiments.

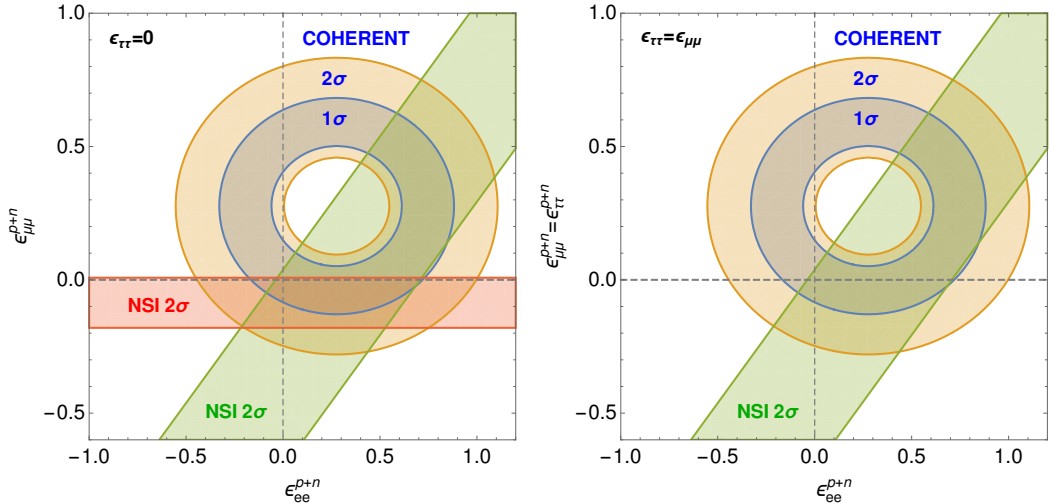

Figure 1: Allowed regions for diagonal muon- and electron-neutrino NSI coupled to baryon number, assuming $\epsilon_{\tau\tau} = 0$ (left) and $\epsilon_{\tau\tau} = \epsilon_{\mu\mu}$ (right).

## 3 Calculating NSI Operators from $Z'$ Bosons

A particularly popular class of NSI realizations uses new neutral gauge bosons $Z'$ as $t$-channel mediators in neutrino scattering. Here we will derive the general expressions for $\epsilon$ in terms of the $Z'$ couplings and then discuss the simplest possible UV-complete scenarios. In addition to the direct coupling of the new $U(1)'$ gauge boson to SM fermions we will also allow for mixing between the $Z'$ and the $Z$ and start with the most general Lagrangian describing the mixing. The formalism for $Z$–$Z'$ mixing [52, 53] has been frequently discussed in the literature, see for example Refs. [30, 54].[6] The Lagrangian contains a term with the usual SM expressions, the $Z'$ part, and a term describing kinetic and mass mixing:

$$
\begin{aligned}
\mathcal{L}_{\text{SM}} &= -\frac{1}{4}\hat{B}_{\mu\nu}\hat{B}^{\mu\nu} - \frac{1}{4}\hat{W}^a_{\mu\nu}\hat{W}^{a\mu\nu} + \frac{1}{2}\hat{M}^2_Z\hat{Z}_\mu\hat{Z}^\mu - \frac{\hat{e}}{\hat{c}_W}j^\mu_Y\hat{B}_\mu - \frac{\hat{e}}{\hat{s}_W}j^{a\mu}_W\hat{W}^a_\mu\,, \\
\mathcal{L}_{Z'} &= -\frac{1}{4}\hat{Z}'_{\mu\nu}\hat{Z}'^{\mu\nu} + \frac{1}{2}\hat{M}'^2_Z\hat{Z}'_\mu\hat{Z}'^\mu - \hat{g}'j'^\mu\hat{Z}'_\mu\,, \\
\mathcal{L}_{\text{mix}} &= -\frac{\sin\chi}{2}\hat{Z}'^{\mu\nu}\hat{B}_{\mu\nu} + \delta\hat{M}^2\hat{Z}'_\mu\hat{Z}^\mu\,.
\end{aligned}
\tag{9}
$$

Hatted fields indicate here that those fields have neither canonical kinetic nor mass terms. The two Abelian gauge bosons $\hat{B}$ and $\hat{Z}'$ couple to each other via the term $\hat{Z}'^{\mu\nu}\hat{B}_{\mu\nu}$, which induces kinetic mixing of $\hat{Z}'$ with the other gauge bosons [52]. It is allowed by the gauge symmetry and hence should be expected. Even if zero at some scale, this term is generated at loop level if there are particles charged under hypercharge and $U(1)'$ [53]. Tree-level mass mixing via the term $\delta\hat{M}^2\hat{Z}'_\mu\hat{Z}^\mu$ requires that there is a scalar with a nonzero vacuum expectation value (VEV) charged under the SM and $U(1)'$.

The currents are defined as

$$
\begin{aligned}
j^\mu_Y &= -\frac{1}{2}\sum_{\ell=e,\mu,\tau}\left[\overline{L}_\ell\gamma^\mu L_\ell + 2\overline{\ell}_R\gamma^\mu\ell_R\right] + \frac{1}{6}\sum_{\text{quarks}}\left[\overline{Q}_L\gamma^\mu Q_L + 4\overline{u}_R\gamma^\mu u_R - 2\overline{d}_R\gamma^\mu d_R\right], \\
j^{a\mu}_W &= \sum_{\ell=e,\mu,\tau}\overline{L}_\ell\gamma^\mu\frac{\sigma^a}{2}L_\ell + \sum_{\text{quarks}}\overline{Q}_L\gamma^\mu\frac{\sigma^a}{2}Q_L\,,
\end{aligned}
\tag{10}
$$

---

[6]An analysis for $Z$–$Z'$–$Z''$ mixing was performed in Ref. [55].

with the left-handed $SU(2)$-doublets $Q_L$ and $L_\ell$ and the Pauli matrices $\sigma^a$. The final electric current after electroweak symmetry breaking is given as $j_{\rm EM} \equiv j_W^3 + j_Y$ and the weak neutral current is $j_{\rm NC} \equiv 2j_W^3 - 2\hat{s}_W^2 j_{\rm EM}$. The new neutral current $j'$ of the $U(1)'$ is left unspecified here, but has to contain flavor *non-universal* neutrino interactions in order to generate NSI:

$$j'_\mu \supset \sum_{\alpha,\beta} q_{\alpha\beta} \overline{\nu}_\alpha \gamma_\mu P_L \nu_\beta \,, \tag{11}$$

with some flavor-dependent coupling matrix $q \neq \mathbb{1}$. Below we will consider some simple models that lead to such couplings.

After diagonalization, the physical massive gauge bosons $Z_{1,2}$ and the massless photon couple to a linear combination of $j'$, $j_{\rm NC}$ and $j_{\rm EM}$:

$$\mathcal{L}_{\rm int} = -\left(\, e j_{\rm EM}, \quad \frac{e}{2\hat{s}_W \hat{c}_W} j_{\rm NC}, \quad g' j' \,\right) \begin{pmatrix} 1 & a_1 & a_2 \\ 0 & b_1 & b_2 \\ 0 & d_1 & d_2 \end{pmatrix} \begin{pmatrix} A \\ Z_1 \\ Z_2 \end{pmatrix}. \tag{12}$$

Here the entries of the matrix are

$$\begin{aligned}
a_1 &= -\hat{c}_W \sin\xi \tan\chi \,, \\
b_1 &= \cos\xi + \hat{s}_W \sin\xi \tan\chi \,, \\
d_1 &= \frac{\sin\xi}{\cos\chi} \,, \\
a_2 &= -\hat{c}_W \cos\xi \tan\chi \,, \\
b_2 &= \hat{s}_W \cos\xi \tan\chi - \sin\xi \,, \\
d_2 &= \frac{\cos\xi}{\cos\chi} \,.
\end{aligned} \tag{13}$$

The angles $\chi$ and $\xi$ in the above expressions come from diagonalizing the kinetic and the mass terms of the massive gauge bosons $Z$ and $Z'$, respectively. The diagonalization of the mass matrix is achieved via

$$\begin{pmatrix} \cos\xi & \sin\xi \\ -\sin\xi & \cos\xi \end{pmatrix} \begin{pmatrix} a & b \\ b & c \end{pmatrix} \begin{pmatrix} \cos\xi & -\sin\xi \\ \sin\xi & \cos\xi \end{pmatrix} = \begin{pmatrix} M_1^2 & 0 \\ 0 & M_2^2 \end{pmatrix} \equiv \begin{pmatrix} M_Z^2 & 0 \\ 0 & M_{Z'}^2 \end{pmatrix}, \quad (14)$$

where

$$\tan 2\xi = \frac{2b}{a-c} \text{ with } \begin{cases} a = \hat{M}_Z^2 \,, \\ b = \hat{s}_W \tan\chi \hat{M}_Z^2 + \frac{\delta \hat{M}^2}{\cos\chi} \,, \\ c = \frac{1}{\cos^2\chi} \left( \hat{M}_Z^2 \hat{s}_W^2 \sin^2\chi + 2\hat{s}_W \sin\chi \delta \hat{M}^2 + \hat{M}_{Z'}^2 \right). \end{cases} \tag{15}$$

At energies $E \ll M_{1,2}$, one can integrate out the $Z_1$ and $Z_2$ bosons to obtain the following effective operators:

$$\mathcal{L}_{\rm eff} = -\sum_{i=1,2} \frac{1}{2M_i^2} \left( e j_{\rm EM} a_i + \frac{e}{2\hat{s}_W \hat{c}_W} j_{\rm NC} b_i + g' j' d_i \right)^2. \tag{16}$$

If more $Z'$ bosons are present, the sum would extend over all their mass states [55]. Note that $\hat{s}_W$ reduces to the known weak angle $\sin\theta_W$ for small $Z$–$Z'$ mixing angle $\xi$ [54].

Comparing the effective Lagrangian from Eq. (16) with the NSI operators in Eqs. (3,4) gives from the mixed $j'$–$j_{\rm EM}$ and $j'$–$j_{\rm NC}$ terms the following NSI coefficients for coupling to

electrons, up- and down-quarks:

$$
\epsilon^e_{\alpha\beta} = \sum_{i=1,2} q_{\alpha\beta} \frac{g' d_i}{\sqrt{2} M_i^2 G_F} \left( -e a_i + \frac{e b_i}{2 s_W c_W} \left( -\frac{1}{2} + 2 s_W^2 \right) + g' d_i \frac{\partial j'_\alpha}{\partial \bar{e} \gamma_\alpha e} \right),
$$

$$
\epsilon^u_{\alpha\beta} = \sum_{i=1,2} q_{\alpha\beta} \frac{g' d_i}{\sqrt{2} M_i^2 G_F} \left( \frac{2}{3} e a_i + \frac{e b_i}{2 s_W c_W} \left( \frac{1}{2} - \frac{4}{3} s_W^2 \right) + g' d_i \frac{\partial j'_\alpha}{\partial \bar{u} \gamma_\alpha u} \right), \tag{17}
$$

$$
\epsilon^d_{\alpha\beta} = \sum_{i=1,2} q_{\alpha\beta} \frac{g' d_i}{\sqrt{2} M_i^2 G_F} \left( -\frac{1}{3} e a_i + \frac{e b_i}{2 s_W c_W} \left( -\frac{1}{2} + \frac{2}{3} s_W^2 \right) + g' d_i \frac{\partial j'_\alpha}{\partial \bar{d} \gamma_\alpha d} \right).
$$

The origin of the $a_i$ ($b_i$) terms from the electric and neutral currents is obvious, whereas the $d_i$ terms take into account that the $Z'$ might have direct couplings to matter particles (i.e. first generation charged fermions) even in the absence of $Z$–$Z'$ mixing. Later we will consider cases with and without direct couplings to matter particles.

Forward scattering of neutrinos in matter corresponds to zero momentum exchange, so the above expressions are valid even for very light $Z'$ masses, contrary to e.g. neutrino scattering in COHERENT. Note however that $Z'$ masses below $\sim 5\,\mathrm{MeV}$ are strongly disfavored by cosmology, in particular the number of relativistic degrees of freedom $N_{\mathrm{eff}}$, unless the coupling is made tiny [56–58]. One can still consider minuscule $g'$ and $Z'$ mass with $M_{Z'}/g' \sim 100\,\mathrm{GeV}$ so as to evade $N_{\mathrm{eff}}$ constraints and still have testable NSI [59], but this typically requires an analysis in terms of long-range potentials [60–62] instead of the contact interactions of Eq. (3) and will not be considered here.

## 3.1 NSI without $Z$–$Z'$ mixing

Let us first consider the case of vanishing $Z$–$Z'$ mixing, $\xi = \chi = 0$, which simplifies Eq. (17) substantially. We must then find a $Z'$ that has couplings to matter particles as well as non-universal neutrino couplings. Flavor-violating neutrino couplings $\overline{\nu}_\alpha \not{Z}' P_L \nu_{\beta \neq \alpha}$ are typically difficult to obtain and often, but not always, run into problems with constraints from charged-lepton flavor violation (LFV) [11, 27]. We will therefore focus on flavor-*diagonal* neutrino couplings in the following, which are much easier to obtain. This is also motivated by the recent hints for lepton-flavor non-universality in $B$-meson decays, which can be explained with models that typically give at least diagonal NSI.

There is a very simple class of $Z'$ models that lead to diagonal NSI that will be the focus of this work. We use the fact that, introducing only right-handed neutrinos to the particle content of the SM, the most general anomaly-free $U(1)_X$ symmetry is generated by Eq. (2),

$$
X = r_{BL}(B - L) + r_{\mu\tau}(L_\mu - L_\tau) + r_{\mu e}(L_\mu - L_e)
$$

for arbitrary real coefficients $r_x$ [33] (see also Refs. [34–38]). This gives the current $j'_\alpha = \sum_f X(f) \bar{f} \gamma_\alpha f$, which is vector-like for all charged particles. The first term in Eq. (2) can couple the $Z'$ to matter even in the absence of $Z$–$Z'$ mixing, while the last two terms induce the neutrino-flavor non-universality necessary for NSI, to be discussed below. Aside from being anomaly-free, the above symmetries can also easily accommodate the observed pattern of neutrino masses and mixing. The key point is that one can break the $U(1)_X$ symmetry using only electroweak singlets which then generate a non-trivial right-handed neutrino Majorana mass matrix that leads to the seesaw mechanism [33]. Despite our flavor symmetry we therefore do not have to worry about LFV, as these effects are still heavily suppressed.

Assuming negligible $Z$–$Z'$ mixing, the effective Lagrangian from Eq. (16) becomes very

simple:

$$
\begin{aligned}
\mathcal{L}_{\text{eff}} = & -\frac{(g')^2}{2M_{Z'}^2} j'_\alpha j'^\alpha \\
\supset & -\frac{(g')^2}{M_{Z'}^2} \left[ r_{BL}(\overline{p}\gamma^\alpha p + \overline{n}\gamma^\alpha n) - (r_{BL} + r_{\mu e})\overline{e}\gamma^\alpha e \right] \\
& \times \left[ -(r_{BL} + r_{\mu e})\overline{\nu}_e \gamma_\alpha P_L \nu_e - (r_{BL} - r_{\mu e} - r_{\mu\tau})\overline{\nu}_\mu \gamma_\alpha P_L \nu_\mu - (r_{BL} + r_{\mu\tau})\overline{\nu}_\tau \gamma_\alpha P_L \nu_\tau \right],
\end{aligned}
\tag{18}
$$

where we used the new-physics current generated by Eq. (2) and only kept the terms relevant for NSI. The NSI coefficients with coupling to baryons then take the form

$$
\epsilon_{ee}^{p,n} - \epsilon_{\mu\mu}^{p,n} = -\frac{(g')^2}{2\sqrt{2}G_F M_{Z'}^2} r_{BL}(2r_{\mu e} + r_{\mu\tau}),
\tag{19}
$$

$$
\epsilon_{\tau\tau}^{p,n} - \epsilon_{\mu\mu}^{p,n} = -\frac{(g')^2}{2\sqrt{2}G_F M_{Z'}^2} r_{BL}(2r_{\mu\tau} + r_{\mu e}),
\tag{20}
$$

and similar for those with electrons

$$
\epsilon_{ee}^{e} - \epsilon_{\mu\mu}^{e} = +\frac{(g')^2}{2\sqrt{2}G_F M_{Z'}^2} (r_{BL} + r_{\mu e})(2r_{\mu e} + r_{\mu\tau}),
\tag{21}
$$

$$
\epsilon_{\tau\tau}^{e} - \epsilon_{\mu\mu}^{e} = +\frac{(g')^2}{2\sqrt{2}G_F M_{Z'}^2} (r_{BL} + r_{\mu e})(2r_{\mu\tau} + r_{\mu e}).
\tag{22}
$$

Neutral matter necessarily contains an equal number of protons and electrons, so the relevant combination is actually the sum $\epsilon^p + \epsilon^e$:

$$
(\epsilon_{ee}^{p} + \epsilon_{ee}^{e}) - (\epsilon_{\mu\mu}^{p} + \epsilon_{\mu\mu}^{e}) = +\frac{(g')^2}{2\sqrt{2}G_F M_{Z'}^2} r_{\mu e}(2r_{\mu e} + r_{\mu\tau}),
\tag{23}
$$

$$
(\epsilon_{\tau\tau}^{p} + \epsilon_{\tau\tau}^{e}) - (\epsilon_{\mu\mu}^{p} + \epsilon_{\mu\mu}^{e}) = +\frac{(g')^2}{2\sqrt{2}G_F M_{Z'}^2} r_{\mu e}(2r_{\mu\tau} + r_{\mu e}).
\tag{24}
$$

Non-vanishing NSI in neutrino oscillations without $Z$–$Z'$ mixing thus require either $r_{BL} \neq 0$ in order to generate a coupling to neutrons or $r_{\mu e} \neq 0$ in order to couple to electrons. Naturally, the phenomenology of a $Z'$ depends sensitively on the SM fermions it couples to. In the following we will go through the basic simple coupling structures which arise in this class of $U(1)'$ groups. We first introduce the various experimental probes and then discuss how these compare to the limits on the NSI derived from neutrino oscillations.[7]

Before moving on let us briefly discuss the possibility of realizing the LMA-Dark [63] solution within our $U(1)'$ framework. As is well known, neutrino oscillations in the presence of NSI contain a generalized mass-ordering degeneracy [64–67] that in principle allows for large $\epsilon$ if the neutrino mixing parameters take on different values from the non-NSI LMA scenario. This LMA-Dark region of parameter space requires a large $\epsilon_{ee} - \epsilon_{\mu\mu} = -\mathcal{O}(1)$ but all other NSI much smaller in magnitude, currently compatible with zero [40]. In our $U(1)'$ models the condition $|\epsilon_{\tau\tau} - \epsilon_{\mu\mu}| \ll |\epsilon_{ee} - \epsilon_{\mu\mu}|$ essentially requires that muons and taus carry the same $U(1)'$ charge, which translates into $r_{\mu\tau} = -r_{\mu e}/2$ above. The only non-vanishing NSI are then

$$
(\epsilon_{ee}^{p} + \epsilon_{ee}^{e}) - (\epsilon_{\mu\mu}^{p} + \epsilon_{\mu\mu}^{e}) = +\frac{3(g')^2}{4\sqrt{2}G_F M_{Z'}^2} r_{\mu e}^2,
\tag{25}
$$

$$
\epsilon_{ee}^{n} - \epsilon_{\mu\mu}^{n} = -\frac{3(g')^2}{4\sqrt{2}G_F M_{Z'}^2} r_{\mu e} r_{BL}.
\tag{26}
$$

---

[7]See e.g. Ref. [42] for a discussion of future limits on some of the models under study here.

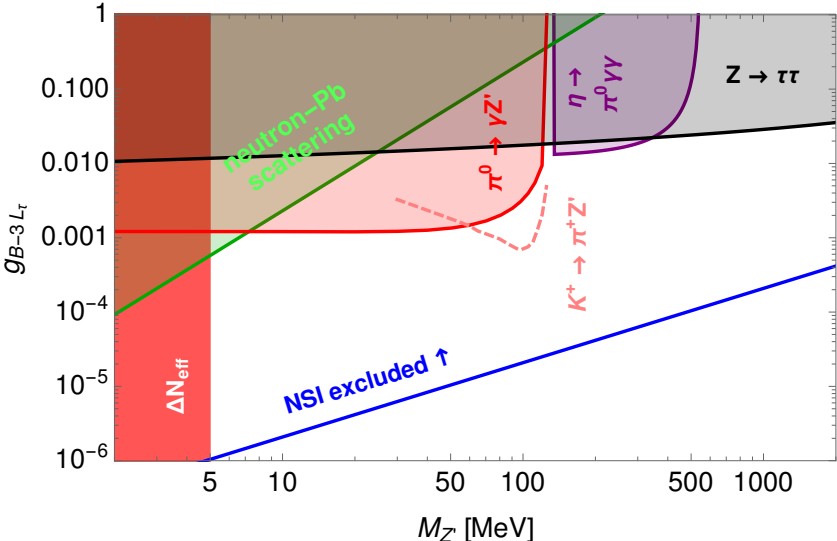

Figure 2: Limits on $U(1)_{B-3L_\tau}$ gauge coupling and $Z'$ mass from Refs. [27,70] together with the strong NSI constraint (blue). For limits that include (radiative) kinetic mixing, see Ref. [71].

The proton plus electron NSI are strictly positive and thus incapable of realizing the LMA-Dark solution; the neutron NSI on the other hand can be negative and even dominant over the proton plus electron NSI by choosing $|r_{\mu e}| \ll |r_{BL}|$. It has however been shown in Ref. [40] that neutron NSI by themselves ($\eta = \pm 90°$ in their notation) do not admit the LMA-Dark solution. This can be easily understood from the highly varying neutron-to-proton density inside the Sun, which explicitly breaks the generalized mass-ordering degeneracy and thus distinguishes between LMA-Dark and LMA [65], the latter providing a significantly better fit [40]. As a result, none of our simple $U(1)'$ models can accommodate the LMA-Dark solution, and so we will not discuss it further. Note that this conclusion remains true if we allow for $Z$–$Z'$ mixing, because this can at best generate neutron NSI as we will see below.

### 3.1.1 Electrophobic NSI

Coming back to the LMA scenario, an interesting special case arises for $r_{\mu e} = -r_{BL} \neq 0$. This assignment of the charges eliminates the coupling to electrons and thus leads to NSI that are generated by the baryon density (i.e. by protons plus neutrons). This simply corresponds to a $U(1)_X$ symmetry generated by $X = B - 2L_\mu - L_\tau + r_{\mu\tau}(L_\mu - L_\tau)$.

Irrespective of the flavor of the leptonic interactions these $U(1)'$ can be probed by purely baryonic processes. In the presence of a light new resonance with a mass below the QCD scale the scattering rates between baryons are modified. The most stringent limits come from measurements of neutron–lead scattering [68,69]. In addition, a light $Z'$ could play a role in meson decays. For $M_{Z'} \lesssim m_{\pi^0}$ the strongest limits come from $\pi^0 \to \gamma + \text{invisible}$, while at higher masses the production of additional hadrons via the $Z'$ can be constrained by a close scrutiny of $\eta$, $\eta'$, $\Psi$ or $\Upsilon$ decays [25]. Limits derived from these observables can be applied to all $U(1)'$ groups that include a coupling to the baryonic current, see for example Fig. 2.

The leptonic couplings of the $Z'$ lead to additional observables which can be used to constrain the interaction strength. On the one hand, couplings to $\tau$ leptons are hard to constrain for $Z'$s in the mass range considered here. The short lifetime and large mass of the $\tau$ prevents a detailed scrutiny of its interaction in low-energy experiments such that we need to rely on the baryonic probes mentioned previously. One of the few relevant $\tau$ constraint comes from

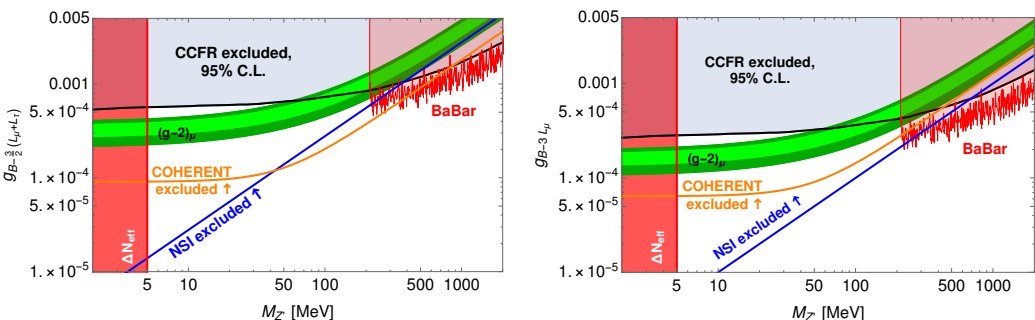

Figure 3: Constraints on $U(1)_{B-\frac{3}{2}(L_\mu+L_\tau)}$ (left) and $U(1)_{B-3L_\mu}$ (right) together with the $2\sigma$ NSI bound from neutrino oscillations (Tab. 2) and the $2\sigma$ constraint from COHERENT. Also shown is the preferred region to resolve the muon's $(g-2)$ at 1 and $2\sigma$ in green and exclusions from $\Delta N_{\text{eff}}$, BaBar [75] and neutrino trident production in CCFR [73,74].

the one-loop vertex correction to the $Z\tau\tau$ and $Z\nu_\tau\nu_\tau$ couplings, which for $M_{Z'} \ll M_Z$ are given by

$$\frac{g_{V,A}}{g_{V,A}^{\text{SM}}} \simeq 1 + \frac{(X(\tau)g')^2}{(4\pi)^2}\left[\frac{\pi^2}{3} - \frac{7}{2} - 3\log\left(\frac{M_{Z'}^2}{M_Z^2}\right) - \log^2\left(\frac{M_{Z'}^2}{M_Z^2}\right) - 3i\pi - 2i\pi\log\left(\frac{M_{Z'}^2}{M_Z^2}\right)\right], \quad (27)$$

with $X(\tau)$ the $U(1)_X$ charge of the tau. The $Z'$ corrections suppress the $Z$ couplings to taus, which have been precisely measured at LEP [72]. We show the naive $2\sigma$ constraint from the axial $Z\tau\tau$ coupling, $|g_A - g_A^{\text{SM}}| < 2 \times 0.00064$ in Fig. 2. While stronger than most $U(1)_B$ limits for $M_{Z'} \sim$ GeV, these limits will not be relevant for $U(1)_X$ models with muon or electron couplings, which are strongly constrained by other observables.

Muons, for example, allow for precision experiments. Rare neutrino-induced processes such as neutrino trident production, which has been measured by the CCFR experiment [73], can test the interaction between neutrinos and muons [74]. As is well known, a light $Z'$ can alleviate the tension between the SM prediction and the measured value of the anomalous magnetic moment of the muon $(g-2)_\mu$. The parameter space in which the tension is reduced to $2\sigma$ ($1\sigma$) is indicated by the dark (light) green band in Fig. 3. In the region above the green band $(g-2)_\mu$ is dominated by the new-physics contribution while $(g-2)_\mu$ asymptotes to the SM value below the green band. Since the new physics can drive the expected anomalous magnetic moment further away from the measurement than the SM a large fraction of the upper region is disfavored compared to the lower regions. We omit this constraint in the figure since this regions is already in tension with CCFR. Additional constraints on a light mediator coupling of muons can be derived from searches for $e^+e^- \to \mu^+\mu^- Z'$ in four-muon final states at BaBar [75]. This search is sensitive down to the two-muon threshold and excludes $g' \gtrsim 10^{-3}$ for $M_{Z'} \simeq 200$ MeV. Finally, there are also constraints from cosmology which are largely insensitive to the details of the particle-physics model. A light $Z'$ can be produced copiously in the early Universe if coupled to light SM fermions, even if just to neutrinos. Bosons with mass below $M_{Z'} \lesssim 5$ MeV then either contribute themselves to the relativistic degrees of freedom $N_{\text{eff}}$ at the time of Big Bang nucleosynthesis [56], or heat up the decoupled neutrino bath via $Z' \to \nu\nu$ [57,58], putting strong constraints on our models.

The relevant NSI limits from a global fit to neutrino oscillation data can be readily read off from Tab. 1. We give the three most extreme cases for $r_{\mu\tau}$ in Tab. 2 which also illustrates the importance of the NSI sign:

- For $B-3L_\tau$ [76–78], corresponding to $r_{\mu\tau} = 2$, we obtain negative NSI coefficients, which are much more constrained than positive NSI. As a result, NSI impose a very strong

Table 2: Examples for NSI from electrophobic anomaly-free $U(1)_X$ without $Z$–$Z'$ mass mixing, as well as the NSI limit [40] on the $Z'$ mass and coupling. See Figs. 2 and 3 for additional limits on the parameter space.

| $U(1)_X$ | $\epsilon_{ee}^{p+n} - \epsilon_{\mu\mu}^{p+n}$ | $\epsilon_{\tau\tau}^{p+n} - \epsilon_{\mu\mu}^{p+n}$ | $M_{Z'}/|g'|$ |
|---|---|---|---|
| $B - 3L_\tau$ | $0$ | $-\frac{3(g')^2}{\sqrt{2}G_F M_{Z'}^2}$ | $> 4.8\,\text{TeV}$ |
| $B - \frac{3}{2}(L_\mu + L_\tau)$ | $+\frac{3(g')^2}{2\sqrt{2}G_F M_{Z'}^2}$ | $0$ | $> 360\,\text{GeV}$ |
| $B - 3L_\mu$ | $+\frac{3(g')^2}{\sqrt{2}G_F M_{Z'}^2}$ | $+\frac{3(g')^2}{\sqrt{2}G_F M_{Z'}^2}$ | $> 1.0\,\text{TeV}$ |

constraint $M_{Z'}/|g'| > 4.8\,\text{TeV}$ on this scenario, to be compared to extremely weak limits from other experiments (see Fig. 2). This is the scenario where neutrino oscillations are most important. COHERENT does not set a limit here because it does not involve tau neutrinos.

- $B - \frac{3}{2}(L_\mu + L_\tau)$ [79], corresponding to $r_{\mu\tau} = 1/2$, gives positive NSI and a rather weak limit of $M_{Z'}/|g'| > 360\,\text{GeV}$. Thanks to the condition $\epsilon_{\tau\tau} = \epsilon_{\mu\mu}$, COHERENT can give better constraints than oscillation data (Fig. 1) and in fact provides the best limit for $40\,\text{MeV} < M_{Z'} < 800\,\text{MeV}$, but is overpowered at higher masses by BaBar [75] and neutrino trident production as measured by CCFR [73,74] (see Fig. 3). At no point can one resolve the longstanding $(g-2)_\mu$ anomaly [80].

- $B - 3L_\mu$ [81], corresponding to $r_{\mu\tau} = -1$, only gives $\epsilon_{\mu\mu}$ and a rather strong limit $M_{Z'}/|g'| > 1\,\text{TeV}$ from neutrino oscillations, which is however weaker than neutrino-trident limits if $M_{Z'} > 700\,\text{MeV}$ (see Fig. 3). As expected from Fig. 1, COHERENT is currently not competitive with oscillation constraints here.

As can be seen, the bounds on hadronic interactions of a $Z'$ are weaker than those arising from interactions with muons. Consequently, we only show the hadronic limits in Fig. 2 and focus on the other constraints in Fig. 3. In all these cases neutrino oscillations provide the strongest limits for light $Z'$, $M_{Z'} = \mathcal{O}(1-100)\,\text{MeV}$, and NSI with a strength that might impair future neutrino oscillation experiments can not be excluded.

### 3.1.2 Electrophilic NSI

Moving on from the electrophobic NSI to $Z'$ scenarios with electron couplings, we again focus on some simple examples to illustrate the different possibilities. Prime examples for relevant $U(1)_X$ generators that lead to $\epsilon^e$ are $B - 3L_e$ [82], $L_e - L_\mu$ [83,84], and $L_e - L_\tau$, collected in Tab. 3.

Models with couplings between neutrinos and electrons allow for additional ways to test the $U(1)'$. First of all, this coupling directly modifies the scattering of neutrinos off electrons. The best limits on the contribution of a light $Z'$ to $\nu$–$e$ scattering come from a reanalysis [41,85] of data collected during the TEXONO-CsI run [86]. In addition, bounds on new interactions with electrons can be derived from positron–electron collisions. The best limits in the mass range of interest here come from the BaBar search for dark photons [87]. When translated into the parameters of the $Z'$ model considered here these limits exclude $g' \gtrsim 10^{-4}$ in a wide range of masses, see e.g. Fig. 4. In addition, there are constraints on light $Z'$ from beam-dump experiments. These bounds can be translated to a given $Z'$ model once the couplings and $Z'$ branching ratios are known [88]. We use the code Darkcast [71] to translate the relevant beam-dump limits [89–95] to the $B - 3L_e$ model, see Fig. 4.

Table 3: Examples for NSI from electrophilic anomaly-free $U(1)_X$ without $Z$–$Z'$ mass mixing, as well as the TEXONO $e$–$\nu$-scattering limit [85] on the $Z'$ mass and coupling and approximate NSI constraints.

| $U(1)_X$ | $\epsilon_{ee}^{e+p} - \epsilon_{\mu\mu}^{e+p}$ | $\epsilon_{ee}^{n} - \epsilon_{\mu\mu}^{n}$ | $M_{Z'}/|g'|$ (TEXONO) | $M_{Z'}/|g'|$ (NSI) |
|---|---|---|---|---|
| $B - 3L_e$ | $+\dfrac{3(g')^2}{\sqrt{2}G_F M_{Z'}^2}$ | $-\dfrac{3(g')^2}{2\sqrt{2}G_F M_{Z'}^2}$ | $> 2\,\text{TeV}$ | $> 0.2\,\text{TeV}$ |
| $U(1)_X$ | $\epsilon_{ee}^{e} - \epsilon_{\mu\mu}^{e}$ | $\epsilon_{\tau\tau}^{e} - \epsilon_{\mu\mu}^{e}$ | $M_{Z'}/|g'|$ (TEXONO) | $M_{Z'}/|g'|$ (NSI) |
| $L_e - L_\mu$ | $+\dfrac{(g')^2}{\sqrt{2}G_F M_{Z'}^2}$ | $+\dfrac{(g')^2}{2\sqrt{2}G_F M_{Z'}^2}$ | $> 0.7\,\text{TeV}$ | $> 0.3\,\text{TeV}$ |
| $L_e - L_\tau$ | $+\dfrac{(g')^2}{2\sqrt{2}G_F M_{Z'}^2}$ | $-\dfrac{(g')^2}{2\sqrt{2}G_F M_{Z'}^2}$ | $> 0.7\,\text{TeV}$ | $> 1.4\,\text{TeV}$ |

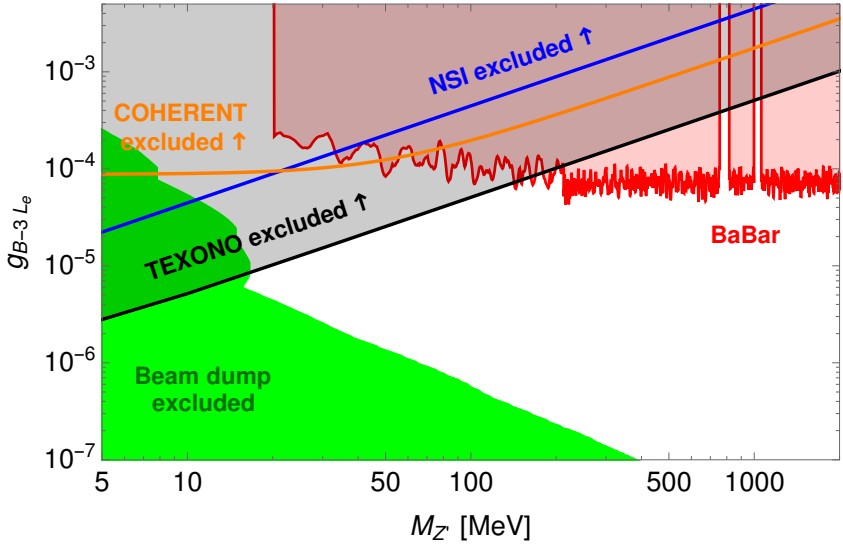

Figure 4: Constraints on $U(1)_{B-3L_e}$ from beam dumps and BaBar (adapted from Refs. [71, 88]) together with COHERENT and TEXONO ($2\sigma$) neutrino scattering bounds [41, 42, 85, 88] as well as approximate NSI constraints.

Since there is no recent analysis of global neutrino oscillation data for NSI that come from the electron density, we have to make some approximations. In principle, the electron matter density and the proton matter density are identical; one is therefore tempted to assume that the limits on proton NSI are the same as those on electron NSI. However, one has to keep in mind that interactions with electrons will not only affect the matter potential (i.e. neutrino propagation) but also the neutrino *detection* process and so bounds of $\epsilon^p$ are not strictly identical to bounds on $\epsilon^e$. Nevertheless, the independent bounds on the interaction of $Z'$ with electrons mentioned above ensure that the neutrino detection process is basically unaffected by new physics. In the following we will hence assume that the limits on proton NSI from the global fit of Ref. [40] are a good proxy for the electron NSI.

Now we can use the limits from Tab. 1 to constrain straightforwardly $L_e - L_{\mu,\tau}$. For $L_e - L_\mu$ the best NSI limit comes from $\epsilon_{\tau\tau}^e - \epsilon_{\mu\mu}^e$ and gives $M_{Z'}/|g'| > 0.3\,\text{TeV}$, a factor of two weaker than the TEXONO limit (Tab. 3). For $L_e - L_\tau$ the best NSI limit also comes from the $\epsilon_{\tau\tau}^e - \epsilon_{\mu\mu}^e$ entry, but is much stronger due to the opposite sign compared to $L_e - L_\mu$; the limit reads $M_{Z'}/|g'| > 1.4\,\text{TeV}$ and is thus a factor two stronger than TEXONO's. This once again illustrates the importance of the NSI sign and the complementarity of the different experiments and observables. Current and future limits in the $M_{Z'}$–$g'$ plane for these two scenarios (without

the NSI bounds) can be found in Ref. [42]. In the last example, $B - 3L_e$, we only generate the $\epsilon_{ee} - \epsilon_{\mu\mu}$ NSI combination, but with contributions from electron, protons, and neutrons of the form $\epsilon^n / \epsilon^{e+p} = -1/2$. Overall this leads to positive $\epsilon_{ee} - \epsilon_{\mu\mu}$ which is then only weakly constrained, $M_{Z'}/|g'| > 0.2\,\text{TeV}$, so that TEXONO is more relevant. We strongly encourage a global analysis of $\epsilon^e$ NSI seeing as they give crucial limits on the parameter space of flavored gauge bosons. Of our three examples, only $B - 3L_e$ can lead to CE$\nu$NS, but this process does not give better limits than TEXONO (Fig. 4).

Going back to the effective Lagrangian (18) one can find another interesting limit around $r_{\mu e} \simeq +r_{BL} \neq 0$, as this would imply a vanishing $\epsilon^p + \epsilon^e + \epsilon^n$ in matter with equal number of protons, neutrons, and electrons. This relation is approximately satisfied inside Earth, which would then be insensitive to this kind of NSI, all the while one could still have large effects in *solar* neutrino oscillations. This corresponds to the case $\eta \simeq -44°$ analyzed in Ref. [40], where it was shown that this scenario indeed severely weakens NSI constraints. Analogously, one can easily imagine a scenario with non-vanishing NSI inside Earth but with $\epsilon \simeq 0$ at one specific radius inside the Sun, once again covered in Ref. [40]. This again weakens the NSI bounds and makes other experimental probes, such as neutrino scattering off electrons and nucleons, more important.

We see again, now more explicitly within UV-complete models, that the flavor structure is crucial to determine which experimental approach can provide the best limits on the model.

## 3.2 NSI with $Z$–$Z'$ mixing

In the cases discussed above, the $Z'$ already had couplings to matter particles $u, d, e$, allowing for NSI without the need for $Z$–$Z'$ mixing. To see the effect of $Z$–$Z'$ mixing, let us consider a simple $U(1)_X$ under which no matter particles are charged. As is obvious from Eq. (2), this singles out $U(1)_{L_\mu - L_\tau}$ [83, 84, 96]. Starting from Eq. (17) it is instructive to obtain the NSI coefficients for protons and neutrons instead of quarks:

$$
\begin{aligned}
\epsilon_{\alpha\beta}^n &= \sum_{i=1,2} q_{\alpha\beta} \frac{e g' d_i}{\sqrt{2} M_i^2 G_F} \frac{b_i}{2 s_W c_W} \left( -\frac{1}{2} \right), \\
\epsilon_{\alpha\beta}^p &= \sum_{i=1,2} q_{\alpha\beta} \frac{e g' d_i}{\sqrt{2} M_i^2 G_F} \left( a_i + \frac{b_i}{2 s_W c_W} \left( \frac{1}{2} - 2 s_W^2 \right) \right), \\
\epsilon_{\alpha\beta}^e &= \sum_{i=1,2} q_{\alpha\beta} \frac{e g' d_i}{\sqrt{2} M_i^2 G_F} \left( -a_i - \frac{b_i}{2 s_W c_W} \left( \frac{1}{2} - 2 s_W^2 \right) \right),
\end{aligned}
\tag{28}
$$

where now $q = \text{diag}(0, 1, -1)$ due to the $U(1)_{L_\mu - L_\tau}$ coupling. Interestingly, proton and electron NSI cancel each other exactly in electrically neutral matter:

$$
\epsilon_{\alpha\beta}^p + \epsilon_{\alpha\beta}^e = 0 \, .
\tag{29}
$$

Note that this result is independent of $L_\mu - L_\tau$, and holds for any $U(1)'$ model one may imagine that has $Z$–$Z'$ mixing but no direct coupling to electrons, up- or down-quarks. Therefore, if the NSI-matter couplings come from $Z$–$Z'$ mixing, the only effects are from coupling to *neutrons* [22], and the limits can be read off Table 1.

Let us take a closer look at the neutron part. An important combination of parameters in the previous expressions is the sum over $b_i d_i / M_i^2$. Using Eqs. (12-14), we can rewrite it as

follows:

$$\sum_{i=1,2} \frac{d_i b_i}{M_i^2} = \frac{1}{c_\chi}\left[ c_\xi s_\xi \left( \frac{1}{M_1^2} - \frac{1}{M_2^2}\right) + s_W t_\chi \left( \frac{s_\xi^2}{M_1^2} + \frac{c_\xi^2}{M_2^2}\right)\right]$$
$$= \frac{\delta\hat{M}^2}{(\delta\hat{M}^2)^2 - \hat{M}_{Z'}^2 \hat{M}_Z^2} = -\frac{\delta\hat{M}^2}{M_1^2 M_2^2 c_\chi^2}. \tag{30}$$

Hence, if there is no, or sufficiently suppressed, mass mixing $\delta\hat{M}^2$, no NSI effects will be generated in neutrino oscillations. In particular, *kinetic mixing* cannot by itself lead to such NSI, even if the $Z'$ has non-universal couplings to neutrinos; *mass mixing* is required, which is a much bigger model-building challenge. Kinetic mixing will of course still lead to effects in neutrino scattering experiments, with the best constraint coming from Borexino [97, 98] rather than COHERENT [99]. Below we will focus on the opposite case where kinetic mixing is absent but mass mixing is present and can thus lead to NSI.

Using Eq. (30), the final NSI for the $L_\mu - L_\tau$ plus mass mixing case are

$$\epsilon_{\tau\tau}^n - \epsilon_{\mu\mu}^n = 2(\epsilon_{ee}^n - \epsilon_{\mu\mu}^n) = -2\frac{eg'}{4\sqrt{2}G_F s_W c_W}\frac{\delta\hat{M}^2}{M_Z^2 M_{Z'}^2 c_\chi^2}, \tag{31}$$

where we denote $M_{1,2} \to M_{Z,Z'}$. These NSI are best constrained by the $\tau\tau - \mu\mu$ NSI: $\epsilon_{\tau\tau}^n - \epsilon_{\mu\mu}^n \in [-0.015, +0.222]$ (see Tab. 1). It is clear from the above expression that the NSI now depend on more parameters of the new physics sector and knowledge of $g'$ and $M_{Z'}$ is no longer sufficient to predict $\epsilon_{\alpha\beta}^n$. Similarly, the neutrino–nucleus scattering cross section tested by COHERENT is sensitive to the $Z$–$Z'$ mixing parameter. As expected from Fig. 1, however, the current COHERENT limit is weaker than the NSI limit due to $\epsilon_{\mu\mu} = -\epsilon_{\tau\tau}$.

Using the (small) $Z$–$Z'$ mixing angle $\xi$ from Eq. (15) the NSI can be expressed as

$$\epsilon_{\tau\tau}^n - \epsilon_{\mu\mu}^n = 2(\epsilon_{ee}^n - \epsilon_{\mu\mu}^n) \simeq -0.04\left(\frac{550\,\mathrm{GeV}}{M_{Z'}/g'}\right)\left(\frac{1\,\mathrm{TeV}}{M_{Z'}/\xi}\right)\left(1 - \frac{M_{Z'}^2}{M_Z^2}\right), \tag{32}$$

showing explicitly that NSI are the result of a cross-coupling of the $L_\mu - L_\tau$ current $g'j'$ and the neutral current $\xi j_{\mathrm{NC}}$. The former is only weakly constrained due to the absence of first-generation particles in $j'$, illustrated in Fig. 5. For light $Z'$, values $M_{Z'}/g' \sim 10\,\mathrm{GeV}$ are possible, whereas heavier $Z'$ are constrained conservatively by CCFR [73] as $M_{Z'}/g' \gtrsim 550\,\mathrm{GeV}$ [74].

The $Z'$ coupling to the non-conserved neutral current $\xi j_{\mathrm{NC}}$ on the other hand gives potentially strong constraints. The most generally applicable bounds are due to additional parity violation and lead to $M_{Z'}/\xi \gtrsim 1\,\mathrm{TeV}$ with little dependence on the details of the UV-completion of the mass-mixing [101–103]. In addition, processes that are sensitive to the emission of the longitudinal $Z'$ are naively expected to receive a $1/M_{Z'}$ enhanced amplitude and, therefore, meson decays such as $K \to \pi Z'$ and $B \to KZ'$ promise strong constraints. However, a certain amount of care is required when dealing with these constraints. In a theory with only mass mixing added to the SM the amplitude is divergent [103]. In the full UV-theory this divergence is canceled by the new physics omitted in the low energy theory and the divergence is replaced by a term $\propto \log(\Lambda^2/M_W^2)$, where $\Lambda$ is the mass scale of the additional degrees of freedom. It has been shown that this estimate reproduces the full result of an exemplary UV-completion well provided that no cancellations occur [103]. In this case $K \to \pi Z'$ gives a limit $M_{Z'}/\xi \gtrsim 10^3\,\mathrm{TeV}$ for $M_{Z'} < 100\,\mathrm{MeV}$ and the CHARM beam-dump gives $\xi < 10^{-8}$ for $\mathrm{MeV} < M_{Z'} < 350\,\mathrm{MeV}$. This indicates that the induced NSI will most likely be severely suppressed for light $Z'$ but we would like to caution that the final answer to this question cannot

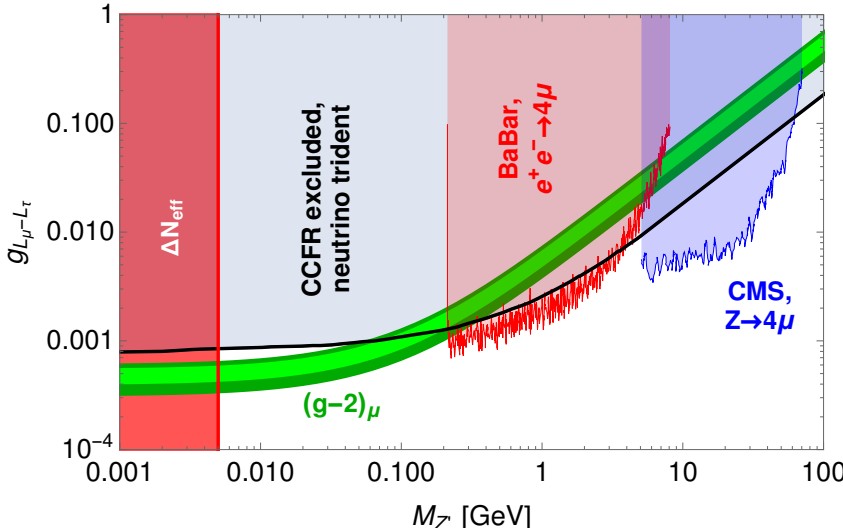

Figure 5: Constraints on $U(1)_{L_\mu-L_\tau}$ without any $Z$–$Z'$ mixing. Shown are the preferred region to resolve the muon's $(g-2)$ at 1 and $2\sigma$ in green and exclusions from $\Delta N_{\text{eff}}$ [57, 58], BaBar [75], CMS [100], and neutrino trident production in CCFR [73, 74].

be given in a model-independent fashion. We note in particular that the $(g-2)_\mu$-motivated region of parameter space cannot give large NSI.

Taken together with the constraints from Fig. 5 we see that the largest NSI in this model can be achieved with a $Z'$ with mass either in the very narrow region around 5 GeV (slightly above the $B \to K Z'$ threshold [103] and below the $Z \to 4\mu$ sensitivity (Fig. 5), although the latter can most likely be pushed down to close this gap) or above $\sim 60$ GeV (above rare-decay thresholds), giving NSI as large as a few percent (Eq. (32)). Depending on the sign of $g'\xi$ this can already be in violation with the global-fit constraints of Tab. 1. However, for such an electroweak-scale $Z'$ above $\sim 60$ GeV one does not just have rare-decay constraints [103] but also direct searches at colliders, e.g. in dilepton channels. From the LHC these are typically only given for $Z'$ masses above 150 GeV (see e.g. Ref. [104]), leaving a gap of currently weakly constrained parameter space [105]. If future neutrino data ever hints at a large $\epsilon^n_{\tau\tau} - \epsilon^n_{\mu\mu}$ then a dedicated search for $\sim 60$–150 GeV-scale $Z'$ would be highly desirable.

As we have seen above, the NSI discussion does not depend on the UV-origin of the $Z$–$Z'$ mass-mixing angle $\xi$, although some of the constraints on $\xi$ do. Let us briefly mention other implications of the UV completion. $Z$–$Z'$ mass mixing unavoidably requires a new scalar that carries both $L_\mu - L_\tau$ and electroweak charge, the simplest example being an additional scalar doublet $\phi'$ with the same hypercharge as the lepton doublet and $L_\mu - L_\tau$ charge $q_{\phi'}$. This gives [30]

$$\delta\hat{M}^2 = \frac{e g' q_{\phi'}}{s_W c_W}\langle\phi'\rangle^2\,,\tag{33}$$

and hence

$$\epsilon^n_{\tau\tau} - \epsilon^n_{\mu\mu} = 2(\epsilon^n_{ee} - \epsilon^n_{\mu\mu}) = -\frac{1}{2\sqrt{2}G_F}\left(\frac{e g'}{s_W c_W}\right)^2 \frac{q_{\phi'}\langle\phi'\rangle^2}{M_Z^2 M_{Z'}^2 c_\chi^2}\,.\tag{34}$$

The vacuum expectation value $\langle\phi'\rangle$ cannot be the only contribution to $M_{Z'}$, so additional electroweak singlets with $L_\mu - L_\tau$ charge are required [22, 106]. The value of $q_{\phi'}$ determines

additional signatures that go beyond the simple $Z$–$Z'$ mass mixing relevant for NSI. For example, in models with $q_{\phi'} = \pm 1$ off-diagonal terms in the charged lepton mass matrix are allowed which induce LFV decays in the sectors $\mu \to e$ (such as $\mu \to e\gamma$, $\mu \to e$ conversion in nuclei) or $\tau \to e$ (such as $\tau \to e\gamma$, $\tau \to 3e$) [22]; in models with $q_{\phi'} = \pm 2$ on the other hand the structure is such that LFV can appear in the tau-mu sector, e.g. in $\tau \to \mu\gamma$ or $h \to \mu\tau$ [106]. Other assignments of $q_{\phi'}$ will not have any impact on LFV and essentially look like a type-I 2HDM. Since these signatures depend additionally on the scalar mixing angle(s) and the scalar mass spectrum, it is difficult to make definite predictions.

# 4 Conclusions

The origin of NSI may be a flavor-sensitive $U(1)'$. Such scenarios face a number of constraints from beam, neutrino scattering and of course oscillation measurements. We demonstrated in this paper that it is quite easy to obtain large *diagonal* NSI in anomaly-free $U(1)'$ models. The models we studied are very well motivated as they are anomaly-free when only right-handed neutrinos are introduced to the particle content of the SM. Neutrino oscillations can often place the strongest constraints on such models if the $Z'$ is in the 10–100 MeV region. These arguably simplest realizations of NSI lead to neutrino scattering off neutrons, protons and electrons in specific combinations.

Some of our key messages may be formulated as follows:

- Large *diagonal* NSI coefficients are possible via a light $Z'$ from an anomaly-free $U(1)_X$ with $X = r_{BL}(B-L) + r_{\mu\tau}(L_\mu - L_\tau) + r_{\mu e}(L_\mu - L_e)$.

- Instead of analyzing NSI for up- and down-quarks one should rather use protons and neutrons as the natural basis.

- The sign of the NSI is fixed by the $U(1)_X$, as is which linear combination of $e$, $p$, and $n$ is relevant for the model. NSI effects in long-baseline experiments can be easily avoided.

- For light $Z'$ one has to carefully distinguish between NSI in oscillations (i.e. forward scattering) and scattering off electrons or nucleons with non-zero momentum transfer.

- NSI and neutrino scattering limits (both $\nu$–$e$ and (coherent) $\nu$–$q$) are complementary and depend strongly on $X$.

- *Kinetic* mixing is not relevant for NSI, but for all other probes.

- If the $U(1)_X$ does not couple to first generation charged fermions, electron and proton NSI cancel each other exactly, and $Z$–$Z'$ *mass* mixing is required to generate effects on neutrons. This mass mixing requires a Higgs multiplet charged under the SM and $U(1)'$ symmetries, and thus in principle testable non-standard Higgs phenomenology.

NSI effects in neutrino oscillations were shown here to be connected to various experimental probes beyond long-baseline or solar neutrino experiments, and surely a broad approach to disentangle their origin will become necessary if any sign of those effects were to be found. On the other hand, well-motivated $Z'$ models were shown to generate NSI effects in oscillations, and should be taken into account when limits on those models are discussed.

## Acknowledgments

We would like to thank Michele Maltoni for providing the values of Tab. 1 and Ivan Esteban for discussions. JH is a postdoctoral researcher of the F.R.S.-FNRS and furthermore supported, in part, by the National Science Foundation under Grant No. PHY-1620638, and by a Feodor Lynen Research Fellowship of the Alexander von Humboldt Foundation. WR is supported by the DFG with grant RO 2516/7-1 in the Heisenberg program.

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
