# Peer review of "Non-Standard Neutrino Interactions and Neutral Gauge Bosons"

_SciPost Physics, doi:SciPost Phys. 6, 038 (2019)_

## Round 1 · Referee Report · Anonymous (Referee 1) · 2018-12-31

Strengths

1- comprehensiveness 2- Well-written

Weaknesses

New idea is not proposed in this paper but the analysis is new and interesting

Report

The article systematically studies anomaly free U(1) gauge theories that lead to neutrino NSI. The article is well-written and contains enough new material to deserve publication. However, the following questions and comments have to be addressed before publication.

Requested changes

1) It should be explained how the ranges associated with $\epsilon^p$, $\epsilon^n$ and $\epsilon^{n+p}$ in Table 1 are related to those for $\epsilon^u$ and $\epsilon^d$.
2) In contrast to the claim in line 224, it is possible to obtain sizeable NSI with light mediator. See e.g. arXiv:1811.01310.
3) Bound on the coupling can be obtained by searching for charged pion and Kaon decay into charged lepton plus neutrino plus $Z’$; i.e., charged lepton plus missing energy. For light $Z’$, it has been shown that the bounds can be very strong. These bounds should be included with a reference to the relevant papers.
4) Line 347: “then” -> “than”
5) In the discussion of “Electrophilic NSI,” The bounds from BOREXINO have to be also included.
6) There are bounds on $\epsilon_{\tau \tau}^e - \epsilon_{\mu \mu}^e$ from long baseline and atmospheric neutrino experiments, see arXiv:1103.4365. Why in the last paragraph of page 13, these bounds are not discussed? Using “proxy” is not very convincing.
7) In the discussion of lines 451-456, the source of LFV has to be clarified.
8) Lines 465 and 466 are misleading. Do the authors refer to the [101,102] results?
9) There are strong bounds on the $Z-Z’$ mass mixing (e.g. from atomic parity violation) which have not been discussed in this article. For a list of these bounds, authors may see arXiv:1402.3620. These bounds should be included.

  • validity: top
  • significance: high
  • originality: good
  • clarity: top
  • formatting: perfect
  • grammar: excellent

Author:  Julian Heeck  on 2019-02-26  [id 451]

(in reply to Report 1 on 2018-12-31)
Category:
answer to question

We thank the referee for careful reading our manuscript and the helpful comments/questions. In the following we explain our response and the corresponding changes to the resubmitted manuscript point by point.

1) As mentioned in the text $\epsilon^p=2 \epsilon^u + \epsilon^d$ and $\epsilon^n= \epsilon^u + 2 \epsilon^d$. The values for $\epsilon^u$ in Tab. 1 are from [40] and correspond to the bound from a global fit with $\epsilon^p=2\epsilon^n$ which has been recast as $\epsilon^u$ with the above relation. Note that there is not direct relation between the values of $\epsilon^{u(d)}$ and $\epsilon^{p(n)}$ in Tab. 1 since these have been extracted from fits with different combinations of the proton and neutron contribution to the matter density. $\epsilon^{u,d}$ are only given in Tab. 1 for reference but are not relevant in the models under consideration.

2) We have never claimed that it was not possible to consider sub-MeV $Z'$ masses, only that the analysis changes. We've added the reference for the interested reader.

3) Pion and Kaon decay bounds can be found in Fig. 2 but are irrelevant both there and for most of the article. The main reason for this is our focus on anomaly-free $U(1)'$ models, which do not have the $\Gamma (\pi,K \to X Z')\propto (g'/m_{Z'})^2$ behavior of anomalous currents that could make meson decays relevant. The only exception is the $U(1){L\mu-L_\tau}$ case with $Z$-$Z'$ mass mixing, which leads to a $Z'$ coupling to the neutral current, in particular to the axial part of it. This then indeed gives rise to constraints from atomic parity violation and meson decays. We have amended our discussion, in particular around and below Eq. (32).

4) Done.

5) For electrophilic NSI, TEXONO gives better limits than Borexino, at least in the $Z'$ mass range under consideration here, see Ref. [85].

6) Long-baseline and atmospheric neutrino data are included in the global fit of arXiv:1805.04530 that we use for our NSI bounds, which supersedes the analysis of arXiv:1103.4365. Our approach of using $\epsilon^p$ limits as limits on $\epsilon^e$ is the best way to approximate NSI limits on $\epsilon^e$ without doing a full analysis that takes detector effects into account. Given that independent bounds on $g'$ and $m_{Z'} $ ensure that detector effects are small we are confident that this approximation is reliable. Furthermore, this is also exactly the way people have treated NSI limits in the past, translating $\epsilon^{e,u,d}$ limits into each other according to their matter density.

7) In models with $|q| \in {1,2}$ off-diagonal contributions to the charged lepton masses are allowed which induce LFV decays. We have clarified this connection in the text. Dedicated analyses of LFV in precisely these types of models can be found in the cited references. The new text, which supersedes the old lines 451-456 reads:

"The value of $q_{\phi'}$ determines additional signatures that go beyond the simple $Z$-$Z'$ mass mixing relevant for NSI. For example, in models with $q_{\phi'} = \pm 1$ off-diagonal terms in the charged lepton mass matrix are allowed which induce LFV decays $\mu \rightarrow e$ or $\tau \rightarrow e$ sector [22]; in models with $q_{\phi'} = \pm 2$ on the other hand the structure is such that LFV can appear in the tau-mu sector, e.g. in $\tau\to \mu\gamma$ or $h\to \mu\tau$ [106]. Other assignments of $q_\phi$ will not have any impact on LFV and essentially look like a type-I 2HDM. Since these signatures depend additionally on the scalar mixing angle(s) and the scalar mass spectrum, it is difficult to make definite predictions"

8) We clarified that sentence by adding the relevant citation to the bound.

9) Indeed, see reply to point 2 of Report 2.

We hope that after these improvements our manuscript is suitable for publication.

---

## Round 1 · Referee Report · Anonymous (Referee 2) · 2019-1-19

Strengths

1) Thoroughness 2) Clarity of exposition

Weaknesses

1) The topic has been discussed before, but the analysis does add new interesting results, in particular those of Sec. 3.2.

Report

In this submission, the authors consider the possibility that nonstandard neutrino interactions originate from the exchange of a flavor-sensitive $Z^\prime$. They focus on a simple class of $U(1)^\prime$ models, which are anomaly-free when three right-handed neutrinos are added to the Standard Model field content. While the subject has been studied before, the paper performs an extensive analysis, presents some new relevant aspects, and is very clearly written. I believe a revised version of the paper will be acceptable for publication, provided the following questions/comments are addressed by the authors.

Requested changes

1) The paper discusses only relatively light $Z^\prime$s, with $M_{Z^\prime} \lesssim 1$ GeV, but the reason for this is not immediately apparent. The authors should make explicit in the text why they did not consider also heavier $Z^\prime$s, which have been invoked for example in relation to the anomalies in $B$ decays (see e.g. Ref. [24]), and are subject to a partially different set of experimental constraints. 2) In the scenario with $Z$-$Z^\prime$ mixing discussed in Sec. 3.2, a priori there can be bounds from $Z$-pole observables and atomic parity violation. Are these relevant? 3) Concerning the bounds in Table 1, it would be helpful if the authors added a short explanation of the asymmetry of the bounds, namely the reason why they are much stronger for negative sign. 4) Lines 75-78: here, I believe at least the prior Ref. [38] (and possibly also earlier work) should be mentioned together with Ref. [33]. 5) In Eq.(9), first line: there should be a factor 1/2 in front of the next-to-last term. 6) In Eq.(12) there should be an overall minus sign, for consistency with Eq.(9). 7) On line 207, the first part of the sentence should be made clearer, replacing it with "At energies $E \ll M_{1,2}$, ..." or similar. 8) In Eq.(19), can the authors double check if the RHS should be multiplied by 2? If so, this also applies to Eqs.(20-26). 9) On line 411, the wording "$U(1)_X$ that does not contain any matter particles" should be improved to "$U(1)_X$ under which no matter particles are charged" or similar. 10) Line 466: towards the end of the line, "bound by" is redundant and should be removed. 11) On line 125, for completeness the relation between $\epsilon_{\alpha\beta}^{p+n}$ and $\epsilon_{\alpha \beta}^{u,d}$ should be given.

  • validity: top
  • significance: high
  • originality: good
  • clarity: top
  • formatting: perfect
  • grammar: excellent

Author:  Julian Heeck  on 2019-02-26  [id 450]

(in reply to Report 2 on 2019-01-19)
Category:
answer to question
reply to objection

We thank the referee for careful reading our manuscript and the helpful comments/questions. In the following we explain our response and the corresponding changes to the resubmitted manuscript point by point.

1) We have focussed on light $Z'$ bosons because these are typically weaker constrained than heavier ones. Neutrino scattering constraints (be it NSI or laboratory) are typically most relevant in the MeV to GeV mass range for $Z'$, as can be seen in many of our figures. Of course the limits derived in our article remain valid for higher $Z'$ masses since they simply constrain the ratio ${m_{Z'}}/{g'}$.

2) The thank the referee for pointing us towards these constraints. Atomic parity violation and even more so rare meson decays, e.g. $K\to \pi Z'$ do indeed give strong constraints on the $Z$-$Z'$ mass mixing angle $\xi$. This impacts only the last section of our article and we have included it in the discussion around and below Eq. (32).

3) The asymmetry in the NSI bounds, i.e. that negative $\epsilon_{ee,\tau\tau}-\epsilon_{\mu\mu}$ are stronger constrained than positive $\epsilon_{ee,\tau\tau}-\epsilon_{\mu\mu}$, has been a feature of global fits for a long time (see e.g. arXiv:1307.3092), but we are not aware of a physical interpretation.

4) We have added ref. 38 as requested.

5) Corrected the typo by multiplying $j_Y$ in eq. (10) by 1/2.

6) We realized that the previous version of eq. (12) is slightly misleading. We have added $\mathcal{L}= - ...$ to eq. (12) for clarity.

7) Done.

8) We checked to ensure that eq. (19) is indeed correct as written.

9) Done.

10) Done.

11) Done.

We hope that after these improvements our manuscript is suitable for publication.

---

## Round 2 · Referee Report · Anonymous (Referee 2) · 2019-3-2

Report

In v2 of the manuscript and/or the accompanying letter, the authors have addressed all the comments I had raised in my first report. In v2 the discussion of Sec.~3.2 (starting from line 439, and including Fig.~5) has been significantly changed to take into account the constraints from atomic parity violation and rare meson decays that apply when $Z$-$Z^\prime$ mass mixing is present, and had been omitted in v1. Concerning this revised discussion I have two requests, listed below. I believe that, aside from these remaining minor issues, the current manuscript is acceptable for publication.

Requested changes

1) The last equality in Eq. (32) was derived assuming $M_{Z^\prime} \ll M_Z$ (it would be useful to state this explicitly), but then on lines 462-464 the authors apply it when referring to a $Z^\prime$ with mass around $100$ GeV. This should be fixed.
2) There are a few typos: line 451, require -> required; line 456, completions -> completion; line 482, ''LFV decays $\mu \to e$ or $\tau \to e$ sector'' should be clarified; line 484, $q_\phi$ -> $q_{\phi^\prime}$.

---

## Round 2 · Referee Report · Anonymous (Referee 1) · 2019-3-11

Strengths

Clarity

Weaknesses

The replies to criticism in the reports are short and not comprehensive and elaborate.

Report

The authors have addressed the questions raised so I find the article now suitable for publication.

---

## Round 2 · Author Response

We thank the referees for careful reading our manuscript and the helpful comments/questions. Our response to the first two reports and the corresponding changes to the manuscript can be found in the comments to the reports. We hope that after these improvements our manuscript is suitable for publication.

---

## Round 3 · Author Response

We thank the referees for the additional comments. We've fixed the typos and modified eq. (32) so that it is valid for arbitrary Z' masses. We've further elaborated the discussion below eq. (32). This should address all the questions and issues brought up by the referees.

---

## Editorial Decision

published